# Effect of Al Addition on Grain Refinement and Phase Transformation of the Mg-Gd-Y-Zn-Mn Alloy Containing LPSO Phase

**DOI:** 10.3390/ma15051632

**Published:** 2022-02-22

**Authors:** Jing Zuo, Mingquan Zhang, Taiki Nakata, Guisong Wang, Danyang Li, Hailong Shi, Chao Xu, Xiaojun Wang, Wenjun Li, Guohua Fan, Lin Geng, Shigeharu Kamado

**Affiliations:** 1Department of Mechanical Engineering, Nagaoka University of Technology, Nagaoka 940-2188, Japan; zuo_jing@126.com (J.Z.); nakata@mech.nagaokaut.ac.jp (T.N.); kamado@mech.nagaokaut.ac.jp (S.K.); 2School of Materials Science and Engineering, Harbin Institute of Technology, Harbin 150001, China; zmq1106435777@163.com (M.Z.); wangguisong@hit.edu.cn (G.W.); lidanyang940120@163.com (D.L.); teleohone@163.com (H.S.); xjwang@hit.edu.cn (X.W.); genglin@hit.edu.cn (L.G.); 3Beijing Institute of Spacecraft Systems Engineering, Beijing 100086, China; 4Key Laboratory for Light-weight Materials, Nanjing Tech University, Nanjing 211816, China; ghfan@hit.edu.cn

**Keywords:** magnesium alloys, rare earth (RE), grain refinement, secondary phases

## Abstract

The effect of 0–1.0 at.% Al additions on grain refinement and phase transformation of the Mg-2.0Gd-1.2Y-0.5Zn-0.2Mn (at.%) alloy containing a long period stacking ordered (LPSO) phase was investigated in this work. The addition of Al promoted the formation of the Al_2_RE phase in the Mg-2.0Gd-1.2Y-0.5Zn-0.2Mn (at.%) alloy, and the dominant secondary phases in the as-cast Mg-2.0Gd-1.2Y-0.5Zn-0.2Mn-1.0Al (at.%) alloy were the Mg_3_RE phase, LPSO phase, and Al_2_RE phase. With increased Al addition, the area fraction of the Al_2_RE phase increased monotonously, while the area fraction of LPSO phase and Mg_3_RE phase decreased gradually. The orientation relationship between the Al_2_RE phase and the α-Mg matrix was determined to be <112>_Al2RE/_/<11$\bar{2}$0>_α-Mg_, {101}_Al2RE/_/{10$\bar{1}$0}_α-Mg_, which was not affected by Zn and Mn concentrations in the Al_2_RE phase. Since the Al_2_RE particles with a size more than 6 μm located at the center of grains could act as nucleants for α-Mg grains, the average grain size of the as-cast alloys decreased from 276 μm to 49 μm after 1.0% Al addition. The effect of the Al addition on the grain refinement of the Mg-2.0Gd-1.2Y-0.5Zn-0.2Mn alloy was comparable to that of the Zr refined counterpart.

## 1. Introduction

Magnesium (Mg) alloys have great application prospects in the automobile and aerospace sectors, by virtue of their low density and high specific strength [1,2,3,4,5]. However, some weaknesses, such as their low strength and poor ductility, greatly limit their extensive applications [6]. It is well known that the addition of rare earth (RE) elements, such as Gd and Y, can significantly improve the mechanical performance of Mg alloys via remarkable solution strengthening and age-hardening responses [7,8]. Meanwhile, the addition of Zn enhances the age-hardening response of the Mg-Gd-Y alloy, and also promotes the formation of the long period stacking ordered (LPSO) phase, which can extremely improve the strength of the alloy [9,10,11]. Thus, a series of works on high strength Mg-Gd-Y-Zn alloys have been reported [12,13,14,15]. For instance, our latest research developed a high strength Mg-11.2Gd-3.8Y-1.2Zn-0.3Mn (wt.%) alloy with a yield strength of about 454 MPa [16]. Thus, Mg-Gd-Y-Zn system alloys are particularly attractive because of their special microstructure and good mechanical properties.

It is universally acknowledged that grain refinement is an effective way to strengthen Mg alloys, due to its constrained slip system at room temperature [17]. Therefore, some grain refiners have been found and applied. For Mg-RE-Zn alloys, Zr is the most effective grain refiner and is normally introduced via binary Mg-Zr master alloys during the fabrication process [18]. However, Zr particles prefer to settle down to the bottom of the crucible more rapidly, due to the density difference between Zr (7.8 g/cm^3^) and Mg (1.7 g/cm^3^) during the melting process, which subsequently leads to the loss of the expensive Zr element and lower utilization of the master alloys [19]. In addition, grain growth occurs easily in the alloy during heating treatment because of the poor thermal stability of Zr particles [20], revealing the disadvantage of Zr acting as grain refiner for Mg-RE-Zn alloys. 

It has been reported that Al addition to the Mg-10Y (wt.%) and Mg-10Gd (wt.%) alloys brings about the formation of the binary thermally stable Al_2_Y and Al_2_Gd Laves phases, respectively, which can act as effective heterogeneous nucleation sites to refine grains during solidification. Moreover, the thermal stability of the grains refined by Al_2_Y and Al_2_Gd phases during high temperature is much higher than those refined by Zr [21,22]. Moreover, the substitution of Al for Zr benefits the cost performance of the alloys. The LPSO phase can also be formed in Mg-RE-Al alloys during the solution process [23], and obviously improve the mechanical properties by short-fiber strengthening [10,11]. It is generally accepted that the strengthening efficiency of the LPSO phase depends on its area fraction and the distribution in the matrix. However, the area fraction of the LPSO phase in Mg-RE-Al alloys is quite low because of the preferential formation of Al_2_RE phase rather than the LPSO phase. Accordingly, in this work, we systematically investigated the effect of Al addition on grain refinement and phase transformation of the as-cast Mg-Gd-Y-Zn-Mn alloy with a high-volume fraction of LPSO phase, which can contribute to developing fine-grained Mg alloys with dense LPSO phases.

## 2. Experimental Procedures

The Mg-2.0Gd-1.2Y-0.5Zn-0.2Mn-xAl (x = 0, 0.2, 0.5, and 1.0 at.%) alloy billets were manufactured by the permanent mold-casting method, High purity Mg (99.89 wt.%) and Mg-30Gd (wt.%), Mg-30Y (wt.%), Mg-30Zn (wt.%) Mg-3Mn (wt.%) Mg-30Al (wt.%) master alloys were melted in a mild steel crucible at 760 °C under the protection of SF_6_ and a CO_2_ mixed gas atmosphere. The melt was poured into a permanent mold at 710 °C. Finally, a cylindric ingot with the length of with 50 mm in diameter and 250 mm in length was fabricated successfully: the alloys are denoted as 0.5Zn-0Al, 0.5Zn-0.2Al, 0.5Zn-0.5Al, 0.5Zn-1.0 Al (all components are expressed as atomic % in this work unless otherwise stated), respectively. For the purpose of analyzing the grain refinement efficiency of the Al addition and the impact of the Zn addition on the LPSO phase formation, the alloys containing Zr and without Zn were also prepared, whose nominal compositions were Mg-2.0Gd-1.2Y-0.5Zn-0.2Mn-0.15Zr and Mg-2.0Gd-1.2Y-0.2Mn-1.0Al, which are denoted as 0.15Zr and 0Zn-1Al, respectively.

The microstructure was characterized by optical microscopy (OM, Olympus BX6000M), a field-emission scanning electron microscope (SEM, JEOL JSM-7000F) equipped with an X-ray energy-dispersive spectrometer (EDS) operating at 20 kV, and transmission electron microscopy (TEM, JEOL JEM-2100F) operating at 200 kV. The thin foils of 0.2 mm thickness used for TEM observation were punched into discs of 3-mm diameter and mechanically polished, followed by low-angle ion milling using the Gatan precision ion-polishing system. X-ray diffraction (XRD) measurements were operated at 40 kV and 15 mA with Cu K_α_ radiation to obtain the phase analysis. The texture was examined by an EDAX-TSL EBSD system operating at 25 kV, and the EBSD data analyses were performed using the TSL OIM 7.0 software. 

## 3. Results

### 3.1. Grain Refinement of as-Cast Alloy by Al Addition

Figure 1 shows OM micrographs of the as-cast alloys. With 0% and 0.2% Al additions, the grains were coarse dendrites and eutectic phases mainly distributed along the dendrite arms (Figure 1a,b). When the Al addition increased to 0.5%, the morphology of grains was completely equiaxed dendritic, with eutectic phases distributed along the dendritic boundaries (Figure 1c). When the concentration of Al increased to 1.0%, the alloy still possessed fully equiaxed dendritic grains, but the cell size became smaller. Besides, some particles were observed in the matrix (Figure 1d). Without Zn addition, the grains remained in equiaxed dendritic morphology, but the cell size slightly increased (Figure 1e). By contrast, the size of the equiaxed grains of the Zr alloy was comparable to that of the 0.5Zn-1Al alloy (Figure 1f), which indicates that the addition of 1.0% Al and 0.15% Zr in the alloy had the same grain refinement effect on the α-Mg matrix. 

Figure 2 shows SEM micrographs of the as-cast alloys with different Al additions. It shows that the microstructure of the alloy was mainly composed of α-Mg grains and eutectic phases, the discontinuous eutectic phases were mainly distributed along the dendrite arms in 0.5Zn-0.2Al and 0.5Zn-0.5Al alloys (Figure 2a,b), with increasing Al concentration to 1.0%, and the morphology of the eutectic phases, mainly distributed along the equiaxed dendritic boundaries, changed into semi-continuous. Moreover, some particles (indicated by yellow arrows) with polygonal morphology gradually preferentially formed at the center of the grains (Figure 2c). This suggests that the formation of the polygonal phase was closely associated with the Al concentration. Figure 3 shows the SEM image and the corresponding EDS elemental maps of Gd, Y, Zn, Al and Mn elements obtained from the eutectic phases in the 0.5Zn-1.0Al alloy. The results show that those polygonal particles distributed in the center of α-Mg grains were enriched with Gd, Y, and Al, with Mn dissolved in it. Similar results were also observed in the as-cast Mg-Y-Al alloy and the polygonal phase was confirmed as the Al_2_Y phase [21]. These polygonal particles that formed in the 0.5Zn-1.0Al and 0Zn-1Al alloys are thought to act as the heterogeneous nucleation sites [22]. Without Zn addition, the eutectic phases were mainly distributed along the equiaxed dendritic boundaries, and some particles were also observed at the center of grains. However, it is noted that some plate-like phases (as circled by red dotted circles) formed in the alloys with Zn addition, but were invisible in the alloy without Zn addition, which means that the formation of such plate-like phases depends on the addition of Zn. According to the corresponding EDS elemental maps (Figure 3b–f), the eutectic phases, including the plate-like phases distributed along the grain boundaries, were enriched with Gd, Y, and Zn. It should be noted that higher element concentration in plate-like phases was observed, in which the Al and Mn atoms segregated.

Figure 4 shows inverse pole figure (IPF) maps of the as-cast alloys. According to the IPF maps, the grains exhibited relatively random orientations and the average grain size decreased from 276 μm to 49 μm, increasing the Al concentration from 0.2% to 1.0%. This suggests that the remarkable grain refinement effect in the 0.5Zn-1.0Al alloy was comparable to that in the alloy with Zr addition (micrographs of the as-cast 0.5Zn-0Al and 0.15Zr alloys have been reported in our previous work) [24]. Without Zn addition, the average grain size was 56 µm, which was slightly coarser than that in the 0.5Zn-1.0Al alloy (Figure 4a–d). Additionally, the area fraction of the Al_2_RE phase increased monotonically with increasing Al additions, that is, it increases gradually from 0.3% for the 0.5Zn-0.2Al alloy to 1.3% for the 0.5Zn-1.0Al alloy, as illustrated in Figure 4e. Meanwhile, the pole figures (PF) exhibited the orientation relationships (ORs) between the labeled Al_2_RE particle and its surrounding α-Mg, as shown in Figure 4f. The ORs were confirmed as {0001}_α-Mg_//{111}_Al2RE_, {10$\bar{1}$0}_α-Mg_//{101}_Al2RE_, {11$\bar{2}$0}_α-Mg_//{112}_Al2RE_. Correlating with the above SEM observation, it can be deduced that only the coarse polygonal Al_2_RE phases with diameters larger than 6 μm, located at the center of grains in the 0.5Zn-1.0Al and 0Zn-1.0Al alloys, can be regarded as efficient heterogeneous nucleation sites. In contrast, the sparse Al_2_RE phases with smaller size in the 0.5Zn-0.2Al and 0.5Zn-0.5Al alloys could not act as effective nucleating agents, leading to their coarser grain size. 

### 3.2. Secondary Phases and Orientation Relationships in as-Cast Alloys

Figure 5 shows XRD patterns obtained from these six alloys. The results suggest that the 0.5Zn-0Al and 0.15Zr alloys without Al addition were mainly composed of Mg_3_RE and LPSO phases (Figure 5a,f). In contrast, besides the Mg_3_RE and LPSO phases, Al addition gave rise to the formation of the Al_2_RE phase in the Al-added alloys (Figure 5b–d), which agrees well with the EBSD analysis. Due to the absence of Zn, the 0Zn-1.0Al alloy mainly consisted of Mg_5_RE and Al_2_RE phases, but without a detectable LPSO phase, demonstrating that Zn rather than Al is the efficient element for the formation of the LPSO phase in Mg-RE alloys (Figure 5e). Furthermore, the Zn addition resulted in the formation of the Mg_3_RE phase rather than the Mg_5_RE phase. Similar results are often observed in Mg-RE-Zn systems [25]. Therefore, the results demonstrate that the co-addition of Al and Zn promoted the formation of the Al_2_RE and LPSO phases in the alloys.

Figure 6 shows the magnified SEM images of the secondary phase morphologies in the as-cast alloys with different Al additions. It shows that there were three main secondary phases based on their morphologies in the alloys with 0.2% and 0.5% Al additions. The phase with brighter contrast is identified as the network-shaped phase and the other two phases with relatively weaker brightness are defined as the plate-like and the lamellae phases, respectively (Figure 6a,b). The morphologies of the secondary phases are closely related to the Al concentrations, and the size of plate-like and network-shaped phases decreased significantly with increasing Al addition (Figure 6c). Moreover, without Zn addition, there was no obvious plate-like phase that formed in the 0Zn-1Al alloy (Figure 6d). The statistic area fractions of the secondary phases versus Al concentration are plotted in Figure 7. It shows that with increasing Al addition, it is obvious that the area fraction of the polygonal phase increased monotonously, while the area fraction of the plate-shaped phase and network-shaped phase decreased gradually.

Figure 8 shows the typical bright-field TEM (BF-TEM) images with the selected area electronic diffraction (SAED) patterns of the secondary phases in the as-cast 0.5Zn-0.5Al alloy. It shows that the particles formed at the grain boundary were comprised of the network-shaped phase (point A) and the plate-like phase (point B) (Figure 8a). Based on the corresponding SAED patterns (Figure 8b,c), these phases are identified as Mg_3_RE (face-centered cubic structure, a = 0.740 nm) and 14H LPSO phases (an ordered hexagonal structure, a = 1.112 nm, c = 3.647 nm) [26,27], respectively. Besides, the polygonal phase (point C) and lamellar structure (point D) (Figure 8d) formed in the grain interior are confirmed as Al_2_RE phase (face-centered cubic structure, a = 0.788 nm) and stacking faults (SFs) [28,29], respectively (Figure 8e,f).

Figure 9 shows the BF-TEM images, atomic-resolution high-angle annular dark field–scanning transmission electron microscopy (HAADF-STEM) micrographs and corresponding SAED patterns of the blocky phases (point E and G) and the plate-like phases (point F and H) in the as-cast 0.5Zn-1.0Al and 0Zn-1.0Al alloys. The results are summarized in Table 1. The results suggest that these phases were Mg_3_RE and 18R LPSO phases (an ordered base-centered monoclinic structure, a = 1.112 nm, b = 1.926 nm, c = 4.689 nm) [27], respectively. The width of the LPSO plate was over 1 μm. Moreover, the 18R LPSO phase was coherent with the Mg_3_RE phase, following a crystallographic OR as [18]_LPSO_//[22]_Mg3RE_ (Figure 9b,c). Meanwhile, the plate-like phase and the blocky phase observed in the as-cast 0Zn-1.0Al alloy are identified as 18R LPSO and Mg_5_RE phases (face-centered cubic structure, a = 2.234 nm) [30], respectively. The width of the LPSO plate was approximately 300 nm. The 18R phase was also coherent with the Mg_5_RE phase, following a crystallographic OR as [18]_LPSO_//[22]_Mg5RE_ (Figure 9e,f). It should be noted that Zn addition has no effect on the orientation relationship. Figure 10a,c show the BF-TEM images of the typical polygonal phases in the as-cast 0.5Zn-1.0Al and 0Zn-1.0Al alloys. According to the corresponding SAED pattern, the polygonal phase with large size was identified as Al_2_RE (Figure 10b). Moreover, the Al_2_RE phase was coherent with the matrix, following a crystallographic OR as [10$\;\bar{1}$0]_ α-Mg_//[110]_Al2RE_, (0002)_ α-Mg_//(111)_Al2RE_ (Figure 10d).

## 4. Discussion

In this work, the addition of Al into the Mg-2.0Gd-1.2Y-0.5Zn-0.2Mn alloy mainly resulted in the formation of Al_2_RE phase, and the number and size of Al_2_RE phases grew with the increase in Al addition. With 0.2% Al addition, the sparse and fine Al_2_RE particles mainly distributed along the dendrite arms, and the average grain size of the 0.5Zn-0.2Al alloy was 276 μm. It has been reported that particles distributed along dendrite arms or grain boundaries cannot act as active heterogeneous nucleation sites [21], and therefore the grains are relatively coarse. As Al concentration increased, more and more Al_2_RE particles formed in the alloys. Especially, some Al_2_RE particles with a size larger than 6 μm were observed at grain centers when 1.0% Al was added in the alloy (Figure 2c), and the average grain size reduced from 276 μm to 49 μm (Figure 4a–d). Due to the small amount of added Al elements, the grain refinement effect caused by the solute element can be ignored. Then, the results suggest that the concentration of Al obviously affects the grain refinement by controlling the distribution and size of the Al_2_RE phase. It has been reported that these particles must be large enough to serve as nucleation sites [31], so these large Al_2_RE particles within grains are more likely to act as active nucleant particles, which might assist the grain refinement.

The grain refinement efficiency of the nucleant particle is associated with crystallographic orientations between the particle and the matrix [32]. The edge-to-edge matching model comprehensively describes the crystallographic characteristics to investigate the atomic matching between two contiguous crystalline phases [33,34]. According to this model, between the particle and the matrix, the potential grain refiner particle definitely has at least one energetically favorable OR, of which the interatomic spacing misfit (f_r_) of one pair of close-packed atomic rows and the interplanar spacing mismatch (f_d_) contain the matching rows less than 10%. The f_r_ and f_d_ values are derived from the following equations:(1)$$f_{r}=│\frac{r_{m}{-r}_{p}}{r_{m}}│\;\times \;100\%$$
(2)$$f_{d}=│\frac{d_{m}{-d}_{p}}{d_{m}}│\;\times \;100\%$$
where d_m_ and d_p_ are the interplanar spacing of matrix and particle, respectively, and r_m_ and r_p_ are the interatomic spacing of matrix and particle, respectively. Smaller values of f_r_ and f_d_ are correlated with higher grain refining efficiency. According to a previous study [21], the matching results can be summarized as the following three ORs:$${<112>}_{\mathrm{Al}2\mathrm{RE}}//{<2\bar{1}\bar{1}0>}_{\alpha -\mathrm{Mg}},\;{\left\{4\bar{4}0\right\}}_{\mathrm{Al}2\mathrm{RE}}//{\left\{0\bar{1}10\right\}}_{\alpha -\mathrm{Mg}}\;\;\;\;\;\;\;\;\;\;\;\;\;\;\;\;\;(\mathrm{OR}(1))$$
$${<112>}_{\mathrm{Al}2\mathrm{RE}}//{<2\bar{1}\bar{1}0>}_{\alpha -\mathrm{Mg}},\;{\left\{\bar{3}31\right\}}_{\mathrm{Al}2\mathrm{RE}}//{\left\{0\bar{1}11\right\}}_{\alpha -\mathrm{Mg}}\;\;\;\;\;\;\;\;\;\;\;\;\;\;\;\;\;(\mathrm{OR}(2))$$
$${<110>}_{\mathrm{Al}2\mathrm{RE}}//{<10\bar{1}0>}_{\alpha -\mathrm{Mg}},\;{\left\{4\bar{4}0\right\}}_{\mathrm{Al}2\mathrm{RE}}//{\left\{0002\right\}}_{\alpha -\mathrm{Mg}}\;\;\;\;\;\;\;\;\;\;\;\;\;\;\;\;\;(\mathrm{OR}(3))$$

Based on the pole figures shown in Figure 4f, the Al_2_RE particle is coherent with the matrix, following a crystallographic OR as {112}_Al2RE_//{11$\bar{2}$0}_α-Mg_, which is equivalent to the direction, i.e., <112>_Al2RE_//<11$\bar{2}$0>_α-Mg_. Therefore, the OR can be expressed as: <112>_Al2RE_//<11$\bar{2}$0>_α-Mg_, {101}_Al2RE_//{10$\bar{1}$0}_α-Mg_. The observed OR between Al_2_RE particle and its surrounding matrix is the same as the predicted OR (1), which is not changed by Zn and Mn concentrations in the Al_2_RE phase. However, according to the TEM results, the OR, i.e., [10$\bar{1}$0]_ α-Mg_//[110]_Al2RE_, (0002)_ α-Mg_//(111)_Al2RE_, is not consistent with the above results, so that the Al_2_RE particle which complies with OR (1) and forms at the center of the grain can be regarded as an active nucleant particle, and contribute to grain refinement for the 0.5Zn-1.0Al alloy.

Although the formation of the LPSO phase has also been reported in Mg-RE-Al alloys, the area fraction is considerably less than that in Mg-RE-Zn alloys with comparable alloying concentrations [23]. This is attributed to the fact that Al prefers to combine with RE to form the polygonal Al_2_RE phase rather than the LPSO phase, as the mixing enthalpy between Al and RE is negatively much larger than the Mg-Al combination [35]. Furthermore, the formation of the Al_2_RE phase further consumes the content of RE in the Mg matrix, leading to a decrease in the required concentration of RE to form the LPSO phase. Thus, the addition of Zn promotes the formation of the LPSO phase in the 0Zn-1.0Al alloy. However, with the increase in Al concentration, the area fraction of the Al_2_RE phase increases monotonously, while that of the LPSO phase and Mg_3_RE phase decrease gradually, and the formation of the Al_2_RE phase leads to a decrease in RE concentration in the alloy. Therefore, the area fraction of the LPSO phase and the Mg_3_RE phase decreases.

## 5. Conclusions

In this study, the effects of Al addition on grain refinement and phase transformation of the Mg-Gd-Y-Zn-Mn alloy containing an LPSO phase was investigated, and the principal conclusions can be summarized as follows:

(1) The main secondary phases in the as-cast 0.5Zn-1.0Al alloy are the Mg_3_RE phase, LPSO phase, and Al_2_RE phase. The addition of Al and Zn promotes the formation of the Al_2_RE phase and the LPSO phase in the alloys. With the increase in Al concentration, the area fraction of the Al_2_RE phase increases monotonously, while the area fraction of LPSO phase and Mg_3_RE phase decreases gradually;

(2) The orientation relationship between the Al_2_RE phase and the α-Mg matrix is determined to be <112>_Al2RE_//<11$\bar{2}$0>_α-Mg_, {101}_Al2RE_//{10$\bar{1}$0}_α-Mg_, which is not affected by Zn and Mn concentrations in the Al_2_RE phase;

(3) The grain size of 0.5Zn-1.0Al alloy containing LPSO phase is remarkably refined by 1.0% Al addition and is comparable to that of alloys with 0.5% Zr addition. Al_2_RE particles with a size more than 6 μm are formed at the center of grains due to the 1.0% Al addition. The average grain size of the as-cast alloy decreases from 276 μm to 49 μm through non-homogeneous nucleation.

## Figures and Tables

**Figure 1 materials-15-01632-f001:**
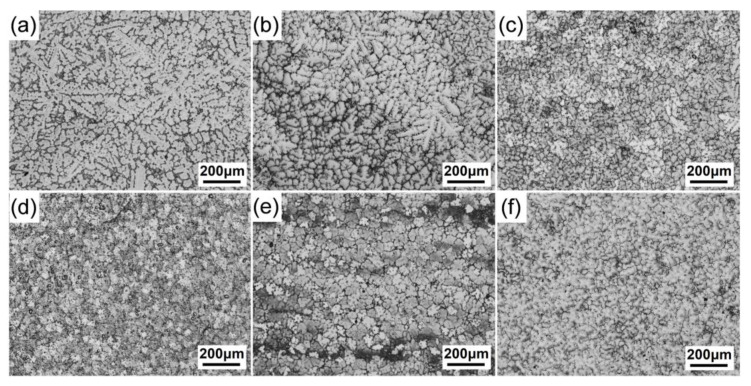
OM micrographs of the as-cast alloys: (**a**–**d**) 0.5Zn-xAl (x = 0, 0.2, 0.5, and 1.0); (**e**) 0Zn-1.0Al and (**f**) 0.15Zr.

**Figure 2 materials-15-01632-f002:**
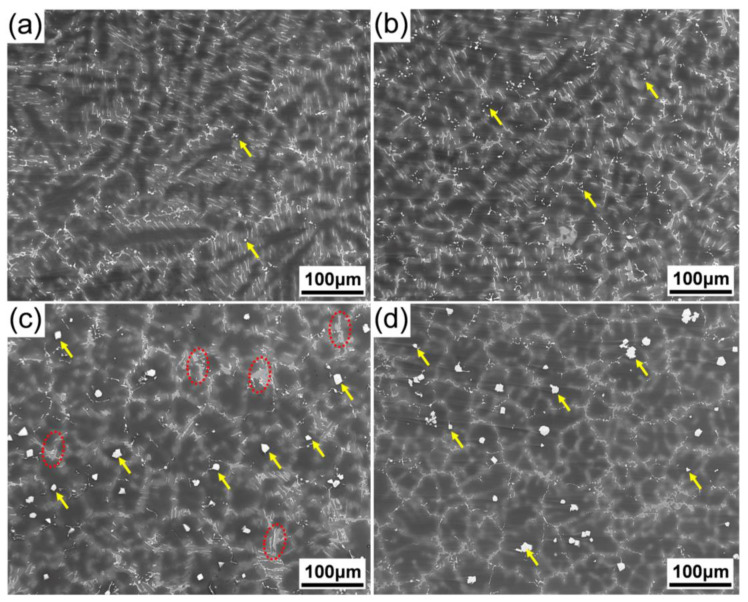
SEM micrographs of the as-cast alloys: (**a**–**c**) 0.5Zn-xAl (x = 0.2, 0.5, and 1.0); (**d**) 0Zn-1.0Al.

**Figure 3 materials-15-01632-f003:**
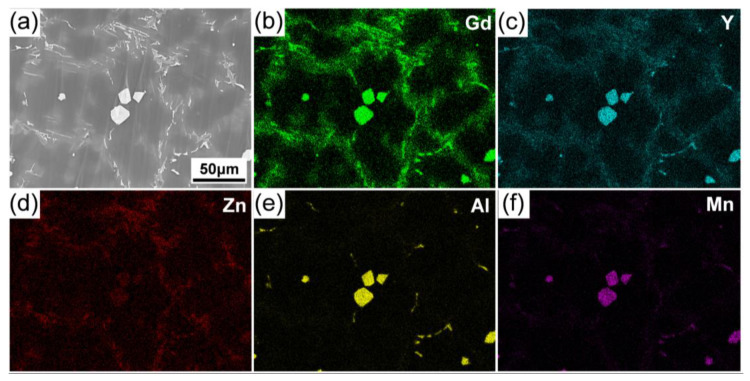
SEM image and EDS elemental maps of eutectic phases in as-cast 0.5Zn-1.0Al alloy: (**a**) SEM image and (**b**–**f**) EDS elemental maps of Gd, Y, Zn, Al and Mn.

**Figure 4 materials-15-01632-f004:**
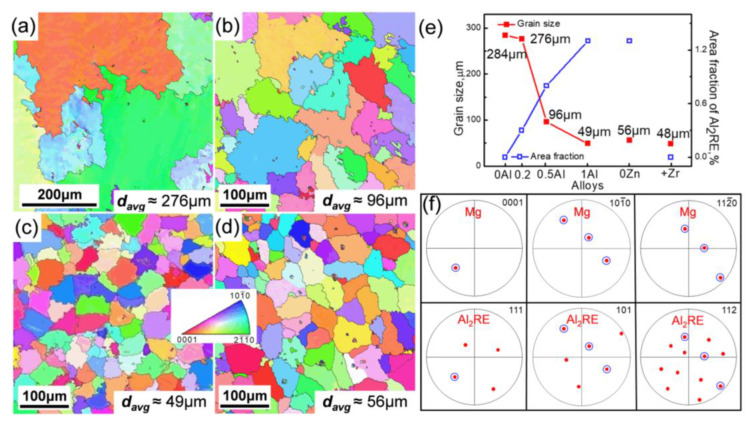
EBSD IPF maps the as-cast alloys: (**a**–**c**) 0.5Zn-xAl (x = 0.2, 0.5, and 1.0); (**d**) 0Zn-1.0Al, (**e**) the line chart about average grain size and Al_2_RE phase area fraction obtained from these six alloys, (**f**) pole figures from both a polygonal particle at a grain center and its surrounding α-Mg.

**Figure 5 materials-15-01632-f005:**
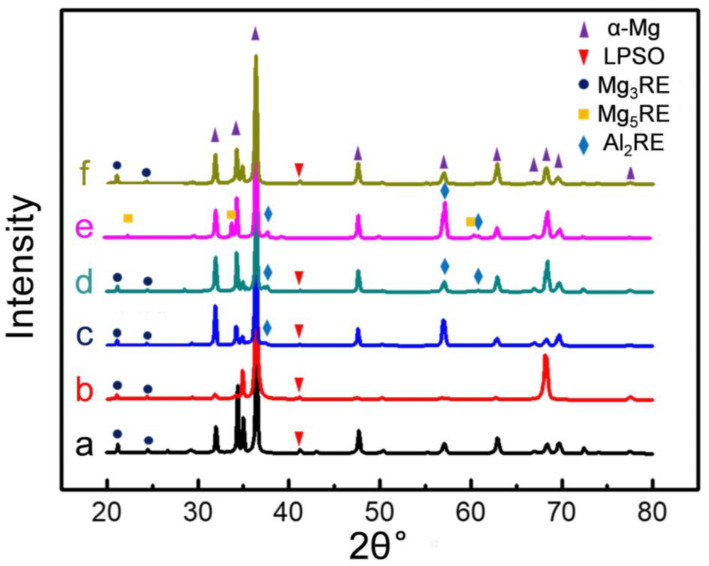
XRD patterns of the as-cast alloys: (**a**–**d**) 0.5Zn-xAl (x = 0, 0.2, 0.5, and 1.0); (**e**) 0Zn-1.0Al and (**f**) 0.15Zr.

**Figure 6 materials-15-01632-f006:**
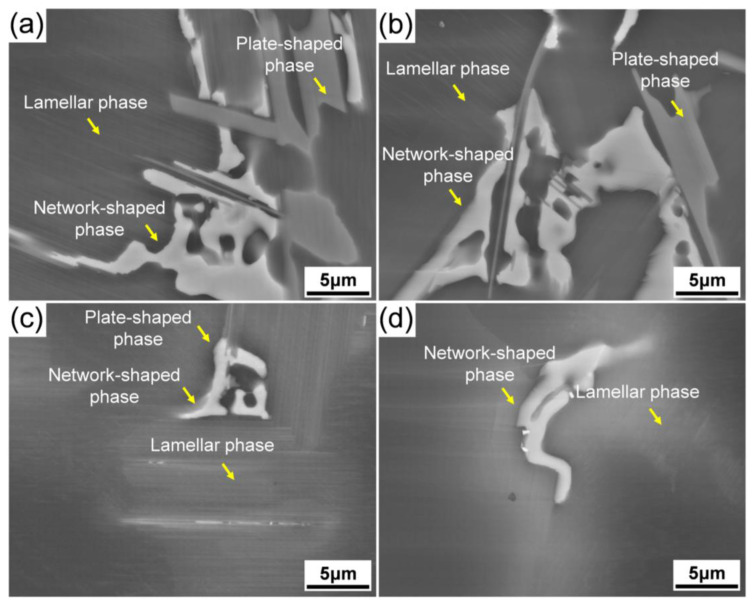
SEM micrographs of the secondary phases in the as-cast alloys: (**a**–**c**) 0.5Zn-xAl (x = 0.2, 0.5, and 1.0); (**d**) 0Zn-1.0Al.

**Figure 7 materials-15-01632-f007:**
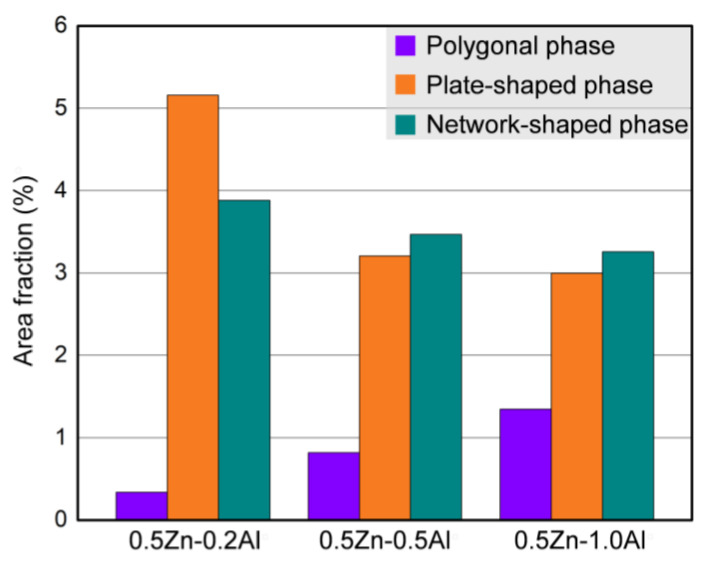
The area fractions of secondary phases in as-cast alloys.

**Figure 8 materials-15-01632-f008:**
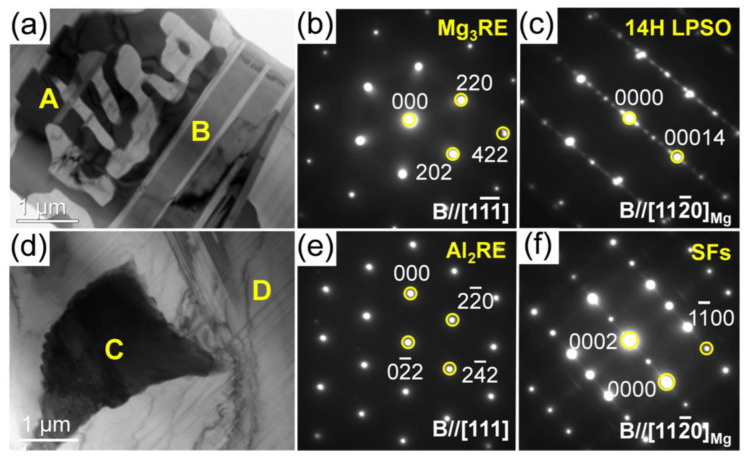
BF-TEM images along with the corresponding SAED patterns of the secondary phases in as-cast 0.5Zn-0.5Al alloy: (**a**,**d**) BF-TEM images; (**b**,**c**,**e**,**f**) the SAED patterns.

**Figure 9 materials-15-01632-f009:**
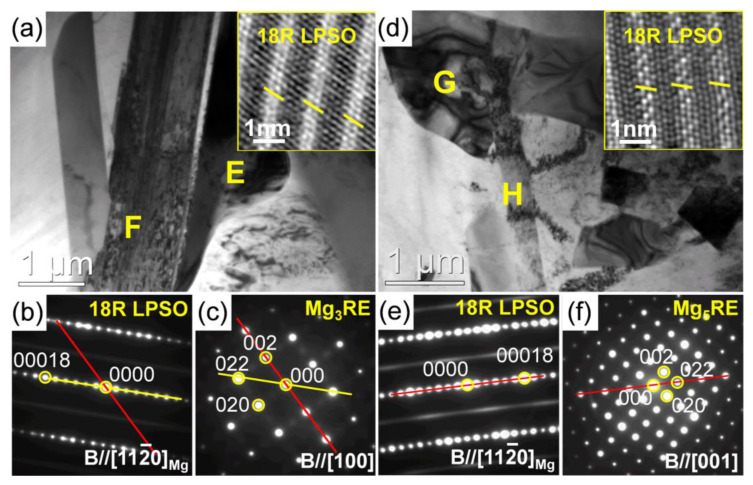
BF-TEM images along with the corresponding SAED patterns of the secondary phases in as-cast alloy: (**a**–**c**) 0.5Zn-1.0Al and (**d**–**f**) 0Zn-1.0Al.

**Figure 10 materials-15-01632-f010:**
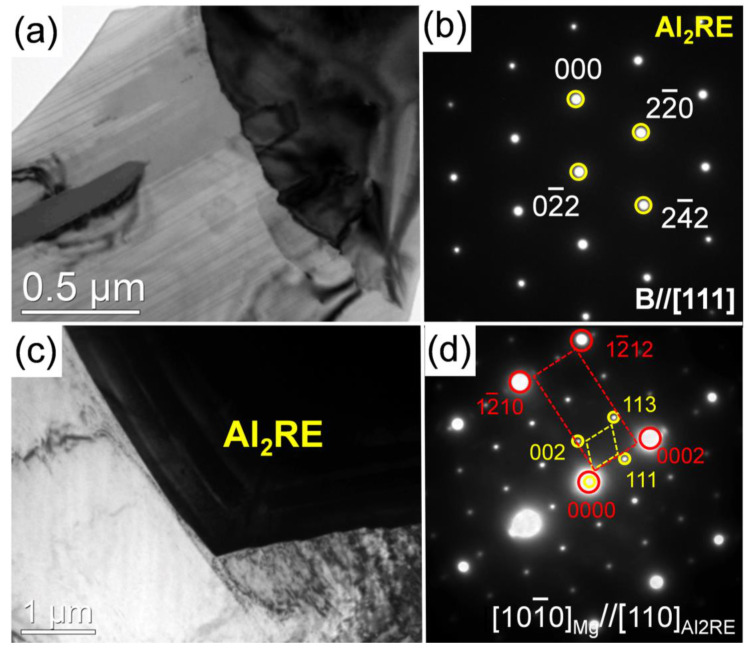
BF-TEM images along with the corresponding SAED patterns of the polygonal phases in as-cast alloy: (**a**,**b**) 0.5Zn-1.0Al and (**c**,**d**) 0Zn-1.0Al.

**Table 1 materials-15-01632-t001:** Summary of the point results of the studied alloys.

Point	A	B	C	D
	Mg_3_RE	14H LPSO	Al_2_RE	SFs
**Point**	**E**	**F**	**G**	**H**
	Mg_3_RE	18R LPSO	Mg_5_RE	18R LPSO

## Data Availability

The data presented in this study are available on request from the corresponding author.

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
