# Peer review of "Effect of Al Addition on Grain Refinement and Phase Transformation of the Mg-Gd-Y-Zn-Mn Alloy Containing LPSO Phase"

_materials, 2022, doi:10.3390/ma15051632_

Round 1
Reviewer 1 Report
Thanks to the authors for the qualitatively presented work, in which a large number of high-quality studies have been carried out.
I have the following questions after reading this article:
1) Describe in more detail the methods for producing alloys. Casting modes, materials used, blank sizes, etc.
2) By what method did the authors confirm the chemical composition of the alloys under study?
3) In figure 3, a phase consisting of 4 elements is clearly visible. (AlMnGdY), did you determine the exact chemical composition of this phase? Has a similar phase been seen before in the literature?
4) Figure 4f. Confidence interval for grain size values?
"Correlating with the above SEM observation, 146 it can be deduced that only the coarse polygonal Al2RE phases with diameters larger than 147 6 μm, located at the center of grains in the 0.5Zn-1.0Al and 0Zn-1.0Al alloys, can be re-148 garded as efficient heterogeneous nucleation sites."
Please, how did you calculate the value of 6 µm?
5) Figure 5. What was the object of study to obtain XRD patterns of the as-cast alloys? Why does the XRD patterns for alloy (b) differ significantly from (a) and (c)? What do you attribute the absence of peaks at ~47, 57, 62 to?
Minor remarks.
Please provide references to the literature using the standard method, e.g. "micrographs of the as-cast 0.5Zn-0Al and 138 0.15Zr alloys have been reported in our previous work) [24]"
Author Response
Response to Reviewer 1 Comments
Thanks to the authors for the qualitatively presented work, in which a large number of high-quality studies have been carried out.
Point 1: Describe in more detail the methods for producing alloys. Casting modes, materials used, blank sizes, etc.
Thanks a lot for the comment and guidance. We have added more detail the methods for producing alloys in the revised paper as follows:
The Mg-2.0Gd-1.2Y-0.5Zn-0.2Mn-xAl (x = 0, 0.2, 0.5, and 1.0 at.%) alloy billets were manufactured by permanent mold casting method, High purity Mg (99.89 wt.%) and Mg-30Gd (wt.%), Mg-30Y (wt.%), Mg-30Zn (wt.%) Mg-3Mn (wt.%) Mg-30Al (wt.%) master alloys were melted in a mild steel crucible at 760 ℃ under the protection of SF6 and CO2 mixed gas atmosphere. The melt was poured into a permanent mold at 710 ℃. Finally, a cylindric ingot with the length of with 50 mm in diameter and 250 mm in length was fabricated successfully.
Point 2: By what method did the authors confirm the chemical composition of the alloys under study?
The chemical composition of the obtained ingot is examined by an inductivity coupled plasma atomic emission spectroscope (ICP-AES).
Point 3: In figure 3, a phase consisting of 4 elements is clearly visible. (AlMnGdY), did you determine the exact chemical composition of this phase? Has a similar phase been seen before in the literature?
These phases you mentioned are enriched with Gd, Y, Al and with Mn dissolved in it. We do not perform the exact chemical composition of this phase, but these phases are identified as Al2RE (Gd, Y) phases and similar result was reported in previous investigation as follow:
J.C. Dai, S.M. Zhu, M. Easton, W.F. Xu, G.H. Wu, W.J. Ding, Precipitation process in a Mg-Gd-Y alloy grain-refined by Al addition, Mater. Charact. 2014, 88, 7-14.
Point 4: Figure 4f. Confidence interval for grain size values?" Correlating with the above SEM observation, 146 it can be deduced that only the coarse polygonal Al2RE phases with diameters larger than 6 μm, located at the center of grains in the 0.5Zn-1.0Al and 0Zn-1.0Al alloys, can be regarded as efficient heterogeneous nucleation sites." Please, how did you calculate the value of 6 µm?
Thanks for the reviewer’s comments. Grain size values were calculated by EBSD analysis, we counted the size of the Al2RE phase in the 0.5Zn-1.0Al and 0Zn-1.0Al alloys using the linear intercept method and found that the size of the Al2RE phase at the centre of the grain is generally larger than 6 μm, the result that was confirmed in other corresponding studies as follow:
G.H. Wu, C.L. Wang, M. Sun, W.J. Ding, Recent developments and applications on high-performance cast magnesium rare-earth alloys, J. Magnes. Alloy. 2021, 9, 1-20.
Point 5: Figure 5. What was the object of study to obtain XRD patterns of the as-cast alloys? Why does the XRD patterns for alloy (b) differ significantly from (a) and (c)? What do you attribute the absence of peaks at ~47, 57, 62 to?
The object of the study to obtain XRD patterns of the as-cast alloy is to illustrate the effect of the addition of Al and Zn elements on phase types in the alloy. For the XRD patterns of alloy (b), the intensity of the diffraction peak produced at ~47, 57, 62 (i.e. Corresponding to the (102), (110) and (103) of a-Mg) is weaker, this may be due to the number of these three crystalline planes involved in diffraction is low in the selected (b) samples due to the difference in orientation, which leads to a decrease in diffraction intensity, but this does not affect our calibration of the a-Mg from the perspective of diffraction peaks.
Minor remarks.
Please provide references to the literature using the standard method, e.g. "micrographs of the as-cast 0.5Zn-0Al and 138 0.15Zr alloys have been reported in our previous work) [24]"
We have modified the relevant content in the revised manuscript.

Reviewer 2 Report
This is a very interesting work, with originality and results clearly presented. The information presented is really significant to transport applications. But your text has extraordinary large paragraphs, must consider reducing them for better clarity and comprehension.
Author Response
This is a very interesting work, with originality and results clearly presented. The information presented is really significant to transport applications. But your text has extraordinary large paragraphs, must consider reducing them for better clarity and comprehension.
We have modified the relevant content in the revised manuscript.
Reviewer 3 Report
The article investigates the effect of Al on grain refinement and a comparison is provided with Zr. Microscopy characterisations were performed to understand the grain refinement by Al2RE. Below are a few comments and suggestions that would improve the article further.
- The experimental procedures should include the casting methods, the temperature used, master alloys used and the information related to casting/mould conditions.
- Experimental procedure line, 80 indicates the Zr composition as 0.15Zr, while line 1-2 on page 3 is indicated as 0.5%Zr.
- Page 3, line 129, referring to the EDS mapping, I could not find Mn segregation. Can the authors mark them in the image?
- Line 141- "Al4RE Phase increases monotonically, "is" should be removed
- Page 5, Fig. 4 (e), can be a bar graph because you have other additions excluding Al. Also, the blue line for are a fraction legend is confusing. I thought that was an additional point.
- Page 8- Figure 8 caption should explain all the contents of the figure separately for (a) to (f)
- The discussion completely ignores the role of solute and the growth restriction by solute elements that also contribute to grain refinement. The authors are advised to add coms points regrading this to the dicussion.
Author Response
The article investigates the effect of Al on grain refinement and a comparison is provided with Zr. Microscopy characterisations were performed to understand the grain refinement by Al2RE. Below are a few comments and suggestions that would improve the article further.
Point 1: The experimental procedures should include the casting methods, the temperature used, master alloys used and the information related to casting/mould conditions.
Thanks a lot for the comment and guidance. We have added more detail the methods for producing alloys in the revised paper as follows.
The Mg-2.0Gd-1.2Y-0.5Zn-0.2Mn-xAl (x = 0, 0.2, 0.5, and 1.0 at.%) alloy billets were manufactured by permanent mold casting method, High purity Mg (99.98 wt.%) and Mg-30Gd (wt.%), Mg-30Y (wt.%), Mg-30Zn (wt.%) Mg-3Mn (wt.%) Mg-30Al (wt.%) master alloys were melted in a mild steel crucible at 760 ℃ under the protection of SF6 and CO2 mixed gas atmosphere. The melt was poured into a permanent mold at 710℃. Finally, a cylindric ingot with the length of with 50 mm in diameter and 250 mm in length was fabricated successfully.
Point 2: Experimental procedure line, 80 indicates the Zr composition as 0.15Zr, while line 1-2 on page 3 is indicated as 0.5%Zr.
Thanks a lot for the guidance. We have modified the relevant content in the revised manuscript.
Point 3: Page 3, line 129, referring to the EDS mapping, I could not find Mn segregation. Can the authors mark them in the image?
Thanks a lot for the guidance. In Fig. 3 (f), the second phase particles located in the center of the grains show purple color, which represents the segregation of Mn.
Point 4: Line 141- "Al2RE Phase increases monotonically, "is" should be removed
Thanks a lot for the guidance. We have modified the relevant content in the revised manuscript.
Point 5: Page 5, Fig. 4 (e), can be a bar graph because you have other additions excluding Al. Also, the blue line for are a fraction legend is confusing. I thought that was an additional point.
Thanks a lot for the guidance. The blue line represents the volume fraction of the Al2RE phase in the alloy. The area fraction of the Al2RE phase increases monotonically with increasing Al addition, that is, increases gradually from 0.3% for the 0.5Zn-0.2Al alloy to 1.3% for the 0.5Zn-1.0Al alloy.
Point 6: Page 8- Figure 8 caption should explain all the contents of the figure separately for (a) to (f)
Thanks a lot for the guidance. We have modified the relevant content in the revised manuscript.
Point 7: The discussion completely ignores the role of solute and the growth restriction by solute elements that also contribute to grain refinement. The authors are advised to add coms points regrading this to the discussion.
Thanks a lot for the comment and guidance. Since the addition of Al element mainly leads to the formation of Al2RE phase located in the center of the grain, and the addition of Al element is small (i.e. 0-1 at.%), so the grain refinement effect caused by the remaining solute element can be ignored, so the main reason for grain refinement is attributed to these large Al2RE particles within grains to act as active nucleant particles.

Reviewer 4 Report
This manuscript of Al addition in Mg-Gd-Y-Zn-Mn alloys is well presented and can be published after following modification.
Authors should add a table summarizing their results with literature reports.
Author Response
This manuscript of Al addition in Mg-Gd-Y-Zn-Mn alloys is well presented and can be published after following modification.
Authors should add a table summarizing their results with literature reports
Thanks a lot for the guidance. We have modified the relevant content in the revised manuscript.
Round 2
Reviewer 1 Report
Accept in present form